# Adversarial GLUE: A Multi-Task Benchmark for Robustness Evaluation of Language Models

[*]**Boxin Wang**[1], [*]**Chejian Xu**[2], **Shuohang Wang**[3], **Zhe Gan**[3],
**Yu Cheng**[3], **Jianfeng Gao**[3], **Ahmed Hassan Awadallah**[3], **Bo Li**[1]
[1]University of Illinois at Urbana-Champaign
[2]Zhejiang University, [3]Microsoft Corporation
{boxinw2,lbo}@illinois.edu, xuchejian@zju.edu.cn
{shuohang.wang,zhe.gan,yu.cheng,jfgao,hassanam}@microsoft.com

## Abstract

Large-scale pre-trained language models have achieved tremendous success across a wide range of natural language understanding (NLU) tasks, even surpassing human performance. However, recent studies reveal that the robustness of these models can be challenged by carefully crafted textual adversarial examples. While several individual datasets have been proposed to evaluate model robustness, a principled and comprehensive benchmark is still missing. In this paper, we present Adversarial GLUE (AdvGLUE), a new multi-task benchmark to quantitatively and thoroughly explore and evaluate the vulnerabilities of modern large-scale language models under various types of adversarial attacks. In particular, we systematically apply 14 textual adversarial attack methods to GLUE tasks to construct AdvGLUE, which is further validated by humans for reliable annotations. Our findings are summarized as follows. ($i$) Most existing adversarial attack algorithms are prone to generating invalid or ambiguous adversarial examples, with around $90\%$ of them either changing the original semantic meanings or misleading human annotators as well. Therefore, we perform careful filtering process to curate a high-quality benchmark. ($ii$) All the language models and robust training methods we tested perform poorly on AdvGLUE, with scores lagging far behind the benign accuracy. We hope our work will motivate the development of new adversarial attacks that are more stealthy and semantic-preserving, as well as new robust language models against sophisticated adversarial attacks. AdvGLUE is available at https://adversarialglue.github.io.

## 1 Introduction

Pre-trained language models [8, 31, 26, 55, 18, 60, 23, 6] have achieved state-of-the-art performance over a wide range of Natural Language Understanding (NLU) tasks [49, 48, 21, 45, 38]. However, recent studies [24, 57, 50, 29, 13] reveal that even these large-scale language models are vulnerable to carefully crafted adversarial examples, which can fool the models to output arbitrarily wrong answers by perturbing input sentences in a human-imperceptible way. Real-world systems built upon these vulnerable models can be misled in ways that would have profound security concerns [27, 28].

To address this challenge, various methods [23, 61, 51, 30] have been proposed to improve the adversarial robustness of language models. However, the adversary setup considered in these methods lacks a unified standard. For example, Jiang et al. [23], Liu et al. [30] mainly evaluate their robustness against human-crafted adversarial datasets [38, 21], while Wang et al. [51] evaluate the

---

[*]Equal Contribution

model robustness against automatic adversarial attack algorithms [24]. The absence of a principled adversarial benchmark makes it difficult to compare the robustness across different models and identify the adversarial attacks that most models are vulnerable to. This motivates us to build a unified and principled robustness evaluation benchmark for natural language models and hope to help answer the following questions: *what types of language models are more robust when evaluated on the unified adversarial benchmark? Which adversarial attack algorithms against language models are more effective, transferable, or stealthy to human? How likely can human be fooled by different adversarial attacks?*

We list out the fundamental principles to create a high-quality robustness evaluation benchmark as follows. First, as also pointed out by [2], a reliable benchmark should be accurately and unambiguously annotated by humans. This is especially crucial for the robustness evaluation, as some adversarial examples generated by automatic attack algorithms can fool humans as well [34]. Given our analysis in §3.4, among the generated adversarial data, there are only around $10\%$ adversarial examples that receive at least 4-vote consensus among 5 annotators and align with the original label. Thus, additional rounds of human filtering are critical to validate the quality of the generated adversarial attack data. Second, a comprehensive robustness evaluation benchmark should cover enough language phenomena and generate a systematic diagnostic report to understand and analyze the vulnerabilities of language models. Finally, a robustness evaluation benchmark needs to be challenging and unveil the biases shared across different models.

In this paper, we introduce Adversarial GLUE (AdvGLUE), a multi-task benchmark for robustness evaluation of language models. Compared to existing adversarial datasets, there are several contributions that render AdvGLUE a unique and valuable asset to the community.

- **Comprehensive Coverage.** We consider textual adversarial attacks from different perspectives and hierarchies, including word-level transformations, sentence-level manipulations, and human-written adversarial examples, so that AdvGLUE is able to cover as many adversarial linguistic phenomena as possible.

- **Systematic Annotations.** To the best of our knowledge, this is the first work that performs systematic and comprehensive evaluation and annotation over 14 different textual adversarial examples. Concretely, AdvGLUE adopts crowd-sourcing to identify high-quality adversarial data for reliable evaluation.

- **General Compatibility.** To obtain comprehensive understanding of the robustness of language models across different NLU tasks, AdvGLUE covers the widely-used GLUE tasks and creates an adversarial version of the GLUE benchmark to evaluate the robustness of language models.

- **High Transferability and Effectiveness.** AdvGLUE has high adversarial transferability and can effectively attack a wide range of state-of-the-art models. We observe a significant performance drop for models evaluated on AdvGLUE compared with their standard accuracy on GLUE leaderboard. For instance, the average GLUE score of ELECTRA(Large) [6] drops from $93.16$ to $41.69$.

Our contributions are summarized as follows. ($i$) We propose AdvGLUE, a principled and comprehensive benchmark that focuses on robustness evaluation of language models. ($ii$) During the data construction, we provide a thorough analysis and a fair comparison of existing strong adversarial attack algorithms. ($iii$) We present thorough robustness evaluation for existing state-of-the-art language models and defense methods. We hope that AdvGLUE will inspire active research and discussion in the community. More details are available at `https://adversarialglue.github.io`.

## 2   Related Work

Existing robustness evaluation work can be roughly divided into two categories: **Evaluation Toolkits** and **Benchmark Datasets**. ($i$) Evaluation toolkits, including OpenAttack [58], TextAttack [35], TextFlint [17] and Robustness Gym [15], integrate various *ad hoc* input transformations for different tasks and provide programmable APIs to dynamically test model performance. However, it is challenging to guarantee the quality of these input transformations. For example, as reported in [57], the validity of adversarial transformation can be as low as $65.5\%$, which means that more than one third of the adversarial sentences have wrong labels. Such a high percentage of annotation errors could lead to an underestimate of model robustness, making it less qualified to serve as an accurate and reliable benchmark [2]. ($ii$) Benchmark datasets for robustness evaluation create challenging

Table 1: **Statistics of AdvGLUE benchmark**. We apply *all* word-level perturbations (C1=*Embedding-similarity*, C2=*Typos*, C3=*Context-aware*, C4=*Knowledge-guided*, and C5=*Compositions*) to the five GLUE tasks. For sentence-level perturbations, we apply *Syntactic-based perturbations* (C6) to the five GLUE tasks. *Distraction-based perturbations* (C7) are applied to four GLUE tasks without QQP, as they may affect the semantic similarity. For human-crafted examples, we apply *CheckList* (C8) to SST-2, QQP, and QNLI; *StressTest* (C9) and *ANLI* (C10) to MNLI; and *AdvSQuAD* (C11) to QNLI tasks.

| Corpus | Task | \|Train\| (GLUE) | \|Test\| (AdvGLUE) | Word-Level | | | | | Sent.-Level | | Human-Crafted | | | |
|---|---|---|---|---|---|---|---|---|---|---|---|---|---|---|
| | | | | C1 | C2 | C3 | C4 | C5 | C6 | C7 | C8 | C9 | C10 | C11 |
| **SST-2** | sentiment | 67,349 | 1,420 | 204 | 197 | 91 | 175 | 64 | 211 | 320 | 158 | 0 | 0 | 0 |
| **QQP** | paraphrase | 363,846 | 422 | 42 | 151 | 17 | 35 | 75 | 37 | 0 | 65 | 0 | 0 | 0 |
| **QNLI** | NLI/QA | 104,743 | 968 | 73 | 139 | 71 | 98 | 72 | 159 | 219 | 80 | 0 | 0 | 57 |
| **RTE** | NLI | 2,490 | 304 | 43 | 44 | 31 | 27 | 23 | 48 | 88 | 0 | 0 | 0 | 0 |
| **MNLI** | NLI | 392,702 | 1,864 | 69 | 402 | 114 | 161 | 128 | 217 | 386 | 0 | 194 | 193 | 0 |
| **Sum of AdvGLUE test set** | | | 4,978 | 431 | 933 | 324 | 496 | 362 | 672 | 1013 | 303 | 194 | 193 | 57 |

testing cases by using human-crafted templates or rules [45, 43, 36], or adopting a human-and-model-in-the-loop manner to write adversarial examples [38, 25, 1]. While the quality and validity of these adversarial datasets can be well controlled, the scalability and comprehensiveness are limited by the human annotators. For example, template-based methods require linguistic experts to carefully construct reasonable rules for specific tasks, and such templates can be barely transferable to other tasks. Moreover, human annotators tend to complete the writing tasks through minimal efforts and shortcuts [4, 47], which can limit the coverage of various linguistic phenomena.

## 3 Dataset Construction

In this section, we provide an overview of our evaluation tasks, as well as the pipeline of how we construct the benchmark data. During this data construction process, we also compare the effectiveness of different adversarial attack methods, and present several interesting findings.

### 3.1 Overview

**Tasks.** We consider the following five most representative and challenging tasks used in GLUE [49]: Sentiment Analysis (*SST-2*), Duplicate Question Detection (*QQP*), and Natural Language Inference (NLI, including *MNLI, RTE, QNLI*). The detailed explanation for each task can be found in Appendix A.3. Some tasks in GLUE are not included in AdvGLUE, since there are either no well-defined automatic adversarial attacks (*e.g.*, *CoLA*), or insufficient data (*e.g.*, *WNLI*) for the attacks.

**Dataset Statistics and Evaluation Metrics.** AdvGLUE follows the same training data and evaluation metrics as GLUE. In this way, models trained on the GLUE training data can be easily evaluated under IID sampled test sets (GLUE benchmark) or carefully crafted adversarial test sets (AdvGLUE benchmark). Practitioners can understand the model generalization via the GLUE diagnostic test suite and examine the model robustness against different levels of adversarial attacks from the AdvGLUE diagnostic report with only one-time training. Given the same evaluation metrics, model developers can clearly understand the performance gap between models tested in the ideally benign environments and approximately worst-case adversarial scenarios. We present the detailed dataset statistics under various attacks in Table 1. Detailed label distribution and evaluation metrics are in Appendix Table 8.

### 3.2 Adversarial Perturbations

In this section, we detail how we optimize different levels of adversarial perturbations to the benign source samples and collect the raw adversarial data with noisy labels, which will then be carefully filtered by human annotators described in the next section. Specifically, we consider the dev sets of GLUE benchmark as our source samples, upon which we perform different adversarial attacks. For relatively large-scale tasks (QQP, QNLI, MNLI-m/mm), we sample 1,000 cases from the dev sets for efficiency purpose. For the remaining tasks, we consider the whole dev sets as source samples.

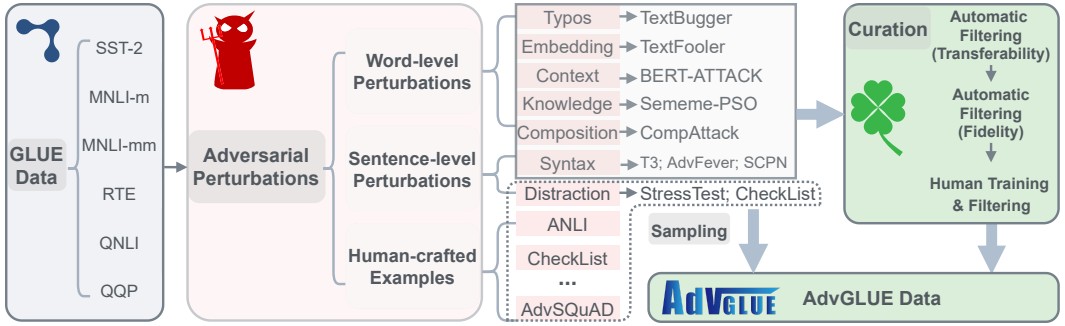

Figure 1: Overview of the AdvGLUE dataset construction pipeline.

### 3.2.1 Word-level Perturbation

Existing word-level adversarial attacks perturb the words through different strategies, such as perturbing words with their synonyms [24] or carefully crafted typo words [27] (*e.g.*, "foolish" to "fo01ish"), such that the perturbation does not change the semantic meaning of the sentences but dramatically change the models' output. To examine the model robustness against different perturbation strategies, we select one representative adversarial attack method for each strategy as follows.

**Typo-based Perturbation.** We select TextBugger [27] as the representative algorithm for generating typo-based adversarial examples. When performing the attack, TextBugger first identifies the important words and then replaces them with typos.

**Embedding-similarity-based Perturbation.** We choose TextFooler [24] as the representative adversarial attack that considers embedding similarity as a constraint to generate semantically consistent adversarial examples. Essentially, TextFooler first performs word importance ranking, and then substitutes those important ones to their synonyms extracted according to the cosine similarity of word embeddings.

**Context-aware Perturbation.** We use BERT-ATTACK [29] to generate context-aware perturbations. The fundamental difference between BERT-ATTACK and TextFooler lies on the word replacement procedure. Specifically, BERT-ATTACK uses the pre-trained BERT to perform masked language prediction to generate contextualized potential word replacements for those crucial words.

**Knowledge-guided Perturbation.** We consider SememePSO [57] as an example to generate adversarial examples guided by the HowNet [41] knowledge base. SememePSO first finds out substitutions for each word in HowNet based on sememes, and then searches for the optimal combination based on particle swarm optimization.

**Compositions of different Perturbations.** We also implement a whitebox-based adversarial attack algorithm called CompAttack that integrates the aforementioned perturbations in one algorithm to evaluate model robustness to various adversarial transformations. Moreover, we efficiently search for perturbations via optimization so that CompAttack can achieve the attack goal while perturbing the minimal number of words. The implementation details can be found in Appendix A.4.

We note that the above adversarial attacks require a surrogate model to search for the optimal perturbations. In our experiments, we follow the setup of ANLI [38] and generate adversarial examples against three different types of models (BERT, RoBERTa, and RoBERTa ensemble) trained on the GLUE benchmark. We then perform one round of filtering to retain those examples with high *adversarial transferability* between these surrogate models. We discuss more implementation details and hyper-parameters of each attack method in Appendix A.4.

### 3.2.2 Sentence-level Perturbation

Different from word-level attacks that perturb specific words, sentence-level attacks mainly focus on the syntactic and logical structures of sentences. Most of them achieve the attack goal by either paraphrasing the sentence, manipulating the syntactic structures, or inserting some unrelated sentences to distract the model attention. AdvGLUE considers the following representative perturbations.

Table 2: **Examples of AdvGLUE benchmark**. We show 3 examples from QNLI task. These examples are generated with three levels of perturbations and they all can successfully change the predictions of all surrogate models (BERT, RoBERTa and RoBERTa ensemble).

| Linguistic Phenomenon | Samples (~~Strikethrough~~ = Original Text, red = Adversarial Perturbation) | Label → Prediction |
|---|---|---|
| Typo (Word-level) | **Question**: What was the population of the Dutch Republic before this emigration? 
 **Sentence**: This was a ~~huge~~ hu ge influx as the entire population of the Dutch Republic amounted to ca. | False → True |
| Distraction (Sent.-level) | **Question**: What was the population of the Dutch Republic before this emigration? https://t.co/DlI9kw 
 **Sentence**: This was a huge influx as the entire population of the Dutch Republic amounted to ca. | False → True |
| CheckList (Human-crafted) | **Question**: What is Tony's profession? 
 **Sentence**: Both Tony and Marilyn were executives, but there was a change in Marilyn, who is now an assistant. | True → False |

**Syntactic-based Perturbation.** We incorporate three adversarial attack strategies that manipulate the sentence based on the syntactic structures. (*i*) *Syntax Tree Transformations*. SCPN [20] is trained to produce a paraphrase of a given sentence with specified syntactic structures. Following the default setting, we select the most frequent 10 templates from ParaNMT-50M corpus [52] to guide the generation process. An LSTM-based encoder-decoder model (SCPN) is used to generate parses of target sentences according to the templates. These parses are further fed into another SCPN to generate full sentences. We use the pre-trained SCPNs released by the official codebase. (*ii*) *Context Vector Transformations*. T3 [50] is a whitebox attack algorithm that can add perturbations on different levels of the syntax tree and generate the adversarial sentence. In our setting, we add perturbations to the context vector of the root node given syntax tree, which is iteratively optimized to construct the adversarial sentence. (*iii*) *Entailment Preserving Transformations*. We follow the entailment preserving rules proposed by AdvFever [45], and transform all the sentences satisfying the templates into semantically equivalent ones. More details can be found in Appendix A.4.

**Distraction-based Perturbation.** We integrate two attack strategies: (*i*) StressTest [36] appends three true statements ("and true is true", "and false is not true", "and true is true" for five times) to the end of the hypothesis sentence for NLI tasks. (*ii*) CheckList [43] adds randomly generated URLs and handles to distract model attention. Since the aforementioned distraction-based perturbations may impact the linguistic acceptability and the understanding of semantic equivalence, we mainly apply these rules to part of the GLUE tasks, including *SST-2* and NLI tasks (*MNLI, RTE, QNLI*), to evaluate whether model can be easily misled by the strong negation words or such lexical similarity.

### 3.2.3 Human-crafted Examples

To ensure our benchmark covers more linguistic phenomena in addition to those provided by automatic attack algorithms, we integrate the following high-quality human-crafted adversarial data from crowd-sourcing or expert-annotated templates and transform them to the formats of GLUE tasks.

**CheckList**[2] [43] is a testing method designed for analysing different capabilities of NLP models using different test types. For each task, CheckList first identifies necessary natural language capabilities a model should have, then designs several test templates to generate test cases at scale. We follow the instructions and collect testing cases for three tasks: *SST-2, QQP* and *QNLI*. For each task, we adopt two capability tests: *Temporal* and *Negation*, which test if the model understands the order of events and if the model is sensitive to negations.

**StressTest**[2] [36] proposes carefully crafted rules to construct "stress tests" and evaluate robustness of NLI models to specific linguistic phenomena. We adopt the test cases focusing on *Numerical Reasoning* into our adversarial *MNLI* dataset. These premise-hypothesis pairs are able to test whether

---

[2]We note that both CheckList and StressTest propose both rule-based distraction sentences and manually crafted templates to generate test samples. The former is considered as sentence-level distraction-based perturbations, while the latter is considered as human-crafted examples.

the model can perform reasoning involving numbers and quantifiers and predict the correct relation between premise and hypothesis.

**ANLI** [38] is a large-scale NLI dataset collected iteratively in a human-in-the-loop manner. In each iteration, human annotators are asked to design sentences to fool current model. Then the model is further finetuned on a larger dataset incorporating these sentences, which leads to a stronger model. Finally, annotators are asked to write harder examples to detect the weakness of this stronger model. In the end, the sentence pairs generated in each round form a comprehensive dataset that aims at examining the vulnerability of NLI models. We adopt ANLI into our adversarial *MNLI* dataset. We obtain the permission from the ANLI authors to include the ANLI dataset as part of our leaderboard.

**AdvSQuAD** [21] is an adversarial dataset targeting at reading comprehension systems. Adversarial examples are generated by appending a distracting sentence to the end of the input paragraph. The distracting sentences are carefully designed to have common words with questions and look like a correct answer to the question. We mainly consider the examples generated by ADDSENT and ADDONESENT strategies, and adopt the distracting sentences and questions in the *QNLI* format with labels "not answered". The use of AdvSQuAD in AdvGLUE is authorized by the authors.

We present sampled AdvGLUE examples with the word-level, sentence-level perturbations and human-crafted samples in Table 2. More examples are provided in Appendix A.5.

### 3.3 Data Curation

After collecting the raw adversarial dataset, additional rounds of filtering are required to guarantee its quality and validity. We consider two types of filtering: automatic filtering and human evaluation.

**Automatic Filtering** mainly evaluates the generated adversarial examples along two fronts: *transferability* and *fidelity*.

1. **Transferability** evaluates whether the adversarial examples generated against one source model (*e.g.*, BERT) can successfully transfer and attack the other two (*e.g.*, RoBERTa and RoBERTa ensemble), given the surrogate models used to generate adversarial examples (BERT, RoBERTa and RoBERTa ensemble). Only adversarial examples that can successfully transfer to the other two models will be kept for the next round of fidelity filtering, so that the selected examples can exploit the biases shared across different models and unveil their fundamental weakness.

2. **Fidelity** evaluates how the generated adversarial examples maintain the original semantics. For word-level adversarial examples, we use *word modification rate* to measure what percentage of words are perturbed. Concretely, word-level adversarial examples with word modification rate larger than $15\%$ are filtered out. For sentence-level adversarial examples, we use *BERTScore* [59] to evaluate the semantic similarity between the adversarial sentences and their corresponding original ones. For each sentence-level attack, adversarial examples with the highest similarity scores are kept to guarantee their semantic closeness to the benign samples.

**Human Evaluation** validates whether the adversarial examples preserve the original labels and whether the labels are highly agreed among annotators. Concretely, we recruit annotators from Amazon Mechanical Turk. To make sure the annotators fully understand the GLUE tasks, each worker is required to pass a training step to be qualified to work on the main filtering tasks for the generated adversarial examples. We tune the pay rate for different tasks, as shown in Appendix Table 11. The pay rate of the main filtering phase is twice as much as that of the training phase.

1. **Human Training Phase** is designed to ensure that the annotators understand the tasks. The annotation instructions for each task follows [37], and we provide at least two examples for each class to help annotators understand the tasks.[3] Each annotator is required to work on a batch of 20 examples randomly sampled from the GLUE dev set. After annotators answer each example, a ground-truth answer will be provided to help them understand whether the answer is correct. Workers who get at least $85\%$ of the examples correct during training are qualified to work on the main filtering task. A total of 100 crowd workers participated in each task, and the number of qualified workers are shown in Appendix Table 11. We also test the human accuracy of qualified annotators for each task on 100 randomly sampled examples from the dev set excluding the training samples. The details and results can be found in Appendix Table 11.

---

[3]Instructions can be found at `https://adversarialglue.github.io/instructions`.

Table 3: **Statistics of data curation**. We report Attack Success Rate (**ASR**) and ASR after data curation (**Curated ASR**) to evaluate the *effectiveness* of different adversarial attacks. We present the **Filter Rate** of data curation and inter-annotator agreement rate (**Fleiss Kappa**) before and after curation to evaluate the *validity* of adversarial examples. **Human Accuracy** on our curated dataset is evaluated by taking one random annotator's annotation as prediction and the majority voted label as ground truth. SPSO: SememePSO, TF: TextFooler, TB:TextBugger, CA: CompAttack, BA:BERT-ATTACK. ↑/↓: higher/lower the better.

| Tasks | Metrics | Word-level Attacks | | | | | Sentence-level Attacks | | | Avg |
| | | SPSO | TF | TB | CA | BA | T3 | SCPN | AdvFever | |
|---|---|---|---|---|---|---|---|---|---|---|
| SST-2 | ASR ↑ | 89.08 | 95.38 | 88.08 | 31.91 | 39.77 | **97.69** | 65.37 | 0.57 | 63.48 |
| | Curated ASR ↑ | 8.29 | 8.97 | 8.85 | 4.02 | 4.04 | **10.45** | 6.88 | 0.23 | 6.47 |
| | Filter Rate ↓ | 90.71 | 90.62 | 90.04 | 86.63 | 89.81 | 89.27 | 89.47 | **60.00** | 85.82 |
| | Fleiss Kappa ↑ | 0.22 | 0.20 | **0.50** | 0.21 | 0.24 | 0.23 | 0.29 | 0.12 | 0.26 |
| | Curated Fleiss Kappa ↑ | 0.51 | 0.49 | **0.67** | 0.46 | 0.45 | 0.44 | 0.47 | 0.20 | 0.52 |
| | Human Accuracy ↑ | 0.85 | 0.86 | **0.91** | 0.88 | 0.85 | 0.78 | 0.85 | 0.50 | 0.87 |
| MNLI | ASR ↑ | 78.45 | 61.50 | 69.35 | 68.58 | 65.02 | **91.23** | 87.73 | 2.25 | 65.51 |
| | Curated ASR ↑ | 3.48 | 1.55 | **8.94** | 3.11 | 2.58 | 3.41 | 6.75 | 0.30 | 3.77 |
| | Filter Rate ↓ | 95.59 | 97.55 | 87.12 | 95.45 | 96.10 | 96.27 | 92.31 | **86.63** | 93.38 |
| | Fleiss Kappa ↑ | 0.28 | 0.24 | **0.53** | 0.39 | 0.32 | 0.28 | 0.24 | 0.35 | 0.33 |
| | Curated Fleiss Kappa ↑ | 0.65 | 0.59 | **0.74** | 0.65 | 0.60 | 0.56 | 0.60 | 0.51 | 0.67 |
| | Human Accuracy ↑ | 0.85 | 0.83 | **0.91** | 0.89 | 0.83 | 0.84 | **0.91** | 0.83 | 0.89 |
| RTE | ASR ↑ | 76.67 | 75.67 | 85.89 | 73.36 | 72.05 | **92.39** | 88.45 | 6.62 | 71.39 |
| | Curated ASR ↑ | 6.20 | 8.14 | **10.03** | 6.97 | 5.58 | 7.05 | 8.30 | 2.53 | 6.85 |
| | Filter Rate ↓ | 91.93 | 89.21 | 88.29 | 90.72 | 92.16 | 92.31 | 90.61 | **61.34** | 87.07 |
| | Fleiss Kappa ↑ | 0.30 | 0.32 | **0.58** | 0.35 | 0.25 | 0.33 | 0.43 | **0.58** | 0.38 |
| | Curated Fleiss Kappa ↑ | 0.49 | 0.67 | **0.80** | 0.63 | 0.42 | 0.60 | 0.64 | 0.65 | 0.66 |
| | Human Accuracy ↑ | 0.77 | **0.95** | 0.94 | 0.87 | 0.79 | 0.89 | 0.91 | 0.86 | 0.92 |
| QNLI | ASR ↑ | 71.88 | 67.03 | 82.54 | 67.24 | 60.53 | **96.41** | 67.37 | 0.97 | 64.25 |
| | Curated ASR ↑ | 3.92 | 2.87 | 5.87 | 4.09 | 2.69 | **7.59** | 3.90 | 0.00 | 3.87 |
| | Filter Rate ↓ | 94.63 | 95.89 | 92.89 | 93.92 | 95.78 | **92.16** | 94.21 | 100.00 | 94.93 |
| | Fleiss Kappa ↑ | 0.07 | 0.05 | **0.16** | 0.10 | 0.14 | 0.07 | 0.12 | -0.16 | 0.11 |
| | Curated Fleiss Kappa ↑ | 0.37 | 0.43 | 0.49 | 0.34 | **0.53** | 0.37 | 0.43 | - | 0.44 |
| | Human Accuracy ↑ | 0.80 | 0.86 | 0.85 | 0.82 | **0.92** | 0.89 | **0.92** | - | 0.85 |
| QQP | ASR ↑ | 45.86 | 48.59 | **57.92** | 49.33 | 43.66 | 48.20 | 44.37 | 0.30 | 42.28 |
| | Curated ASR ↑ | 1.52 | 1.74 | **5.87** | 3.05 | 0.76 | 1.47 | 1.50 | 0.00 | 1.99 |
| | Filter Rate ↓ | 96.73 | 96.50 | **89.90** | 93.83 | 98.28 | 97.04 | 96.62 | 100.00 | 96.11 |
| | Fleiss Kappa ↑ | 0.26 | 0.27 | **0.38** | 0.27 | 0.24 | 0.25 | 0.29 | - | 0.30 |
| | Curated Fleiss Kappa ↑ | 0.32 | 0.46 | **0.62** | 0.48 | 0.40 | 0.10 | 0.47 | - | 0.51 |
| | Human Accuracy ↑ | 0.84 | 0.98 | 0.97 | 0.89 | 0.78 | 0.89 | **1.00** | - | 0.89 |

2. **Human Filtering Phase** verifies the quality of the generated adversarial examples and only maintains high-quality ones to construct the benchmark dataset. Specifically, annotators are required to work on a batch of 10 adversarial examples generated from the same attack method. Every adversarial example will be validated by 5 different annotators. Examples are selected following two criteria: ($i$) high consensus: each example must have at least 4-vote consensus; ($ii$) utility preserving: the majority-voted label must be the same as the original one to make sure the attacks are valid (*i.e.*, cannot fool human) and preserve the semantic content.

The data curation results including inter-annotator agreement rate (Fleiss Kappa) and human accuracy on the curated dataset are shown in Table 3. We will provide more analysis in the next section. Note that even after the data curation step, some grammatical errors and typos can still remain, as some adversarial attacks intentionally inject typos (*e.g.*, TextBugger) or manipulate syntactic trees (*e.g.*, SCPN) which are very stealthy. We will retain these samples as their labels receive high consensus from annotators, which means the typos do not substantially impact humans' understanding.

### 3.4 Benchmark of Adversarial Attack Algorithms

Our data curation phase also serves as a comprehensive benchmark over existing adversarial attack methods, as it provides a fair standard for all adversarial attacks and systematic human annotations to evaluate the quality of the generated samples.

Table 4: **Model performance on AdvGLUE test set**. BERT (Large) and RoBERTa (Large) are fine-tuned using different random seeds and thus different from the surrogate models used for adversarial text generation. For MNLI, we report the test accuracy on the matched and mismatched test sets; for QQP, we report accuracy and F1; and for other tasks, we report the accuracy. All values are reported by percentage (%). We also report the macro-average (Avg) of per-task scores for different models. (Complete results are listed in our leaderboard.)

| Model | SST-2 AdvGLUE | MNLI AdvGLUE | RTE AdvGLUE | QNLI AdvGLUE | QQP AdvGLUE | Avg AdvGLUE | Avg GLUE | Avg $\Delta \downarrow$ |
|---|---|---|---|---|---|---|---|---|
| *State-of-the-art Pre-trained Language Models* | | | | | | | | |
| BERT (Large) | 33.03 | 28.72/27.05 | 40.46 | 39.77 | 37.91/16.56 | 33.68 | 85.76 | 52.08 |
| ELECTRA (Large) | 58.59 | 14.62/20.22 | 23.03 | 57.54 | 61.37/42.40 | 41.69 | **93.16** | 51.47 |
| RoBERTa (Large) | 58.52 | 50.78/39.62 | 45.39 | 52.48 | 57.11/41.80 | 50.21 | 91.44 | 41.23 |
| T5 (Large) | 60.56 | 48.43/38.98 | 62.83 | 57.64 | 63.03/**55.68** | 56.82 | 90.39 | 33.57 |
| ALBERT (XXLarge) | **66.83** | 51.83/44.17 | 73.03 | **63.84** | 56.40/32.35 | 59.22 | 91.87 | 32.65 |
| DeBERTa (Large) | 57.89 | **58.36/52.46** | **78.95** | 57.85 | 60.43/47.98 | **60.86** | 92.67 | **31.81** |
| *Robust Training Methods for Pre-trained Language Models* | | | | | | | | |
| SMART (BERT) | 25.21 | 26.89/23.32 | 38.16 | 34.61 | 36.49/20.24 | 30.29 | 85.70 | 55.41 |
| SMART (RoBERTa) | 50.92 | 45.56/36.07 | 70.39 | 52.17 | **64.22**/44.28 | 53.71 | 92.62 | 38.91 |
| FreeLB (RoBERTa) | 61.69 | 31.59/27.60 | 62.17 | 62.29 | 42.18/31.07 | 50.47 | 92.28 | 41.81 |
| InfoBERT (RoBERTa) | 47.61 | 50.39/41.26 | 39.47 | 54.86 | 49.29/35.54 | 46.04 | 89.06 | 43.02 |

**Evaluation Metrics.** Specifically, we evaluate these attacks along two fronts: *effectiveness* and *validity*. For effectiveness, we consider two evaluation metrics: **Attack Success Rate (ASR)** and **Curated Attack Success Rate (Curated ASR)**. Formally, given a benign dataset $\mathcal{D} = \{(x^{(i)}, y^{(i)})\}_{i=1}^{N}$ consisting of $N$ pairs of sample $x^{(i)}$ and ground truth $y^{(i)}$, for an adversarial attack method $\mathcal{A}$ that generates an adversarial example $\mathcal{A}(x)$ given an input $x$ to attack a surrogate model $f$, ASR is calculated as

$$\textbf{ASR} = \sum_{(x,y)\in\mathcal{D}} \frac{\mathbb{1}[f(\mathcal{A}(x)) \neq y]}{\mathbb{1}[f(x) = y]}, \tag{1}$$

where $\mathbb{1}$ is the indicator function. After the data curation phase, we collect a curated adversarial dataset $\mathcal{D}_c$. Thus, Curated ASR is calculated as

$$\textbf{Curated ASR} = \sum_{(x,y)\in\mathcal{D}} \frac{\mathbb{1}[f(\mathcal{A}(x)) \neq y] \cdot \mathbb{1}[\mathcal{A}(x) \in \mathcal{D}_c]}{\mathbb{1}[f(x) = y]}. \tag{2}$$

For validity, we consider three evaluation metrics: **Filter Rate**, **Fleiss Kappa**, and **Human Accuracy**. Specifically, Filter Rate is calculated by $1 - \frac{\text{Curated ASR}}{\text{ASR}}$ to measure how many examples are rejected in the data curation procedures and can reflect the noisiness of the generated adversarial examples. We report the average ASR, Curated ASR, and Filter Rate over the three surrogate models we consider in Table 3. Fleiss Kappa is a widely used metric in existing datasets (*e.g.*, SNLI, ANLI, and FEVER [3, 38, 46]) to measure the inter-annotator agreement rate on the collected dataset. Fleiss Kappa between 0.4 and 0.6 is considered as moderate agreement and between 0.6 and 0.8 as substantial agreement. The inter-annotator agreement rates of most high-quality datasets fall into these two intervals. In this paper, we follow the standard protocol and report Fleiss Kappa and Curated Fleiss Kappa to analyze the inter-annotator agreement rate on the collected adversarial dataset before and after curation to reflect the ambiguity of generated examples. We also estimate the human performance on our curated datasets. Specifically, given a sample with 5 annotations, we take one random annotator's annotation as the prediction and the majority voted label as the ground truth and calculate the human accuracy as shown in Table 3.

**Analysis.** As shown in Table 3, in terms of attack *effectiveness*, while most attacks show high ASR, the Curated ASR is always less than 11%, which indicates that most existing adversarial attack algorithms are not effective enough to generate high-quality adversarial examples. In terms of *validity*, the filter rates for most adversarial attack methods are more than 85%, which suggests that existing strong adversarial attacks are prone to generating invalid adversarial examples that either change the original semantic meanings or generate ambiguous perturbations that hinder the annotators' unanimity. We provide detailed filter rates for automatic filtering and human evaluation in Appendix Table 12, and the conclusion is that around $60 - 80\%$ of examples are filtered due to the low transferability

Table 5: **Diagnostic report of state-of-the-art language models and robust training methods**. For each attack method, we evaluate models against generated adversarial data for different tasks to obtain per-task accuracy scores, and report the macro-average of those scores. (C1=*Embedding-similarity*, C2=*Typos*, C3=*Context-aware*, C4=*Knowledge-guided*, C5=*Compositions*, C6=*Syntactic-based Perturbations*, C7=*Distraction-based Perturbations*, C8=*CheckList*, C9=*StressTest*, C10=*ANLI* and C11=*AdvSQuAD*).

| Models | Word-Level Perturbations | | | | | Sent.-Level | | Human-Crafted Examples | | | |
|---|---|---|---|---|---|---|---|---|---|---|---|
| | C1 | C2 | C3 | C4 | C5 | C6 | C7 | C8 | C9 | C10 | C11 |
| BERT (Large) | 42.02 | 31.96 | 45.18 | 45.86 | 33.85 | 44.86 | 24.16 | 16.33 | 23.20 | 13.47 | 10.53 |
| ELECTRA (Large) | 43.07 | 45.12 | 47.95 | 46.33 | 47.33 | 43.47 | 33.30 | 32.20 | 26.29 | 26.94 | 52.63 |
| RoBERTa (Large) | 56.54 | 57.19 | 60.47 | 49.81 | 55.92 | 50.49 | 41.89 | 37.78 | 28.35 | 16.58 | 35.09 |
| T5 (Large) | 60.04 | 67.94 | 64.60 | 59.84 | 58.50 | 50.54 | 42.20 | **69.02** | 23.20 | 17.10 | 52.63 |
| ALBERT (XXLarge) | **66.71** | 67.61 | **73.49** | 70.36 | 59.52 | **63.76** | 49.14 | 45.55 | 39.69 | 26.94 | 43.86 |
| DeBERTa (Large) | 65.07 | **74.87** | 68.02 | 65.30 | **62.54** | 57.41 | 47.22 | 45.08 | **52.06** | 22.80 | 54.39 |
| SMART (BERT) | 45.17 | 31.04 | 42.89 | 45.23 | 30.76 | 40.74 | 16.62 | 8.20 | 18.56 | 10.36 | 1.75 |
| SMART (RoBERTa) | 62.93 | 58.03 | 65.09 | 62.65 | 61.37 | 55.31 | 40.13 | 39.27 | 28.35 | 15.54 | 31.58 |
| FreeLB (RoBERTa) | 51.95 | 53.23 | 52.92 | 51.15 | 52.18 | 50.75 | 37.72 | 66.87 | 23.71 | **29.02** | **64.91** |
| InfoBERT (RoBERTa) | 55.47 | 55.78 | 59.02 | 51.33 | 55.48 | 44.56 | 31.49 | 34.31 | 42.27 | 14.51 | 43.86 |

and high word modification rate. Among the remaining samples, around $30 - 40\%$ examples are filtered due to the low human agreement rates (Human Consensus Filtering), and around $20 - 30\%$ are filtered due to the semantic changes which lead to the label changes (Utility Preserving Filtering). We also note that the data curation procedures are indispensable for the adversarial evaluation, as the Fleiss Kappa before curation is very low, suggesting that a lot of adversarial sentences have unreliable labels and thus tend to underestimate the model robustness against the textual adversarial attacks. After the data curation, our AdvGLUE shows a Curated Fleiss Kappa of near 0.6, comparable with existing high-quality dataset such as SNLI and ANLI. Among all the existing attack methods, we observe that TextBugger is the most effective and valid attack method, as it demonstrates the highest Curated ASR and Curated Fleiss Kappa across different tasks.

### 3.5 Finalizing the Dataset

The full pipeline of constructing AdvGLUE is summarized in Figure 1.

**Merging.** We note that distraction-based adversarial examples and human-crafted adversarial examples are guaranteed to be valid by definition or crowd-sourcing annotations, and thus data curation is not needed on these attacks. When merging them with our curated set, we calculate the average number of samples per attack from our curated set, and sample the same amount of adversarial examples from these attacks following the same label distribution. This way, each attack contributes to similar amount of adversarial data, so that AdvGLUE can evaluate models against different types of attacks with similar weights and provide a comprehensive and unbiased diagnostic report.

**Dev-Test Split.** After collecting the adversarial examples from the considered attacks, we split the final dataset into a dev set and a test set. In particular, we first randomly split the benign data into $9 : 1$, and the adversarial examples generated based on $90\%$ of the benign data serve as the hidden test set, while the others are published as the dev set. For human-crafted adversarial examples, since they are not generated based on the benign GLUE data, we randomly select $90\%$ of the data as the test set, and the remaining $10\%$ as the dev set. The dev set is publicly released to help participants to understand the tasks and the data format. To protect the integrity of our test data, the test set will not be released to the public. Instead, participants are required to upload the model to CodaLab, which automates the evaluation process on the hidden test set and provides a diagnostic report.

## 4 Diagnostic Report for Language Models

**Benchmark Results.** We follow the official implementations and training scripts of pre-trained language models to reproduce results on GLUE and test these models on AdvGLUE. The training details can be found in Appendix A.6. Results are summarized in Table 4. We observe that although state-of-the-art language models have achieved high performance on GLUE, they are vulnerable to various adversarial attacks. For instance, the performance gap can be as large as $55\%$ on the SMART

(BERT) model in terms of the average score. DeBERTa (Large) and ALBERT (XXLarge) achieve the highest average AdvGLUE scores among all the tested language models. This result is also aligned with the ANLI leaderboard[4], which shows that ALBERT (XXLarge) is the most robust to human-crafted adversarial NLI dataset [38].

We note that although our adversarial examples are generated from surrogate models based on BERT and RoBERTa, these examples have high transferability between models after our data curation. Specifically, the average score of ELECTRA (Large) on AdvGLUE is even lower than RoBERTa (Large), which demonstrates that AdvGLUE can effectively transfer across models of different architectures and unveil the vulnerabilities shared across multiple models. Moreover, we find some models even perform worse than random guess. For example, the performance of BERT on AdvGLUE for all tasks is lower than random-guess accuracy.

We also benchmark advanced robust training methods to evaluate whether these methods can indeed provide robustness improvement on AdvGLUE and to what extent. We observe that SMART and FreeLB are particularly helpful to improve robustness for RoBERTa. Specifically, SMART (RoBERTa) improves RoBERTa (Large) over $3.71\%$ on average, and it even improves the benign accuracy as well. Since InfoBERT is not evaluated on GLUE, we run InfoBERT with different hyper-parameters and report the best accuracy on benign GLUE dev set and AdvGLUE test set. However, we find that the benign accuracy of InfoBERT (RoBERTa) is still lower than RoBERTa (Large), and similarly for the robust accuracy. These results suggest that existing robust training methods only have incremental robustness improvement, and there is still a long way to go to develop robust models to achieve satisfactory performance on AdvGLUE.

**Diagnostic Report of Model Vulnerabilities.** To have a systematic understanding of which adversarial attacks language models are vulnerable to, we provide a detailed diagnostic report in Table 5. We observe that models are most vulnerable to human-crafted examples, where complex linguistic phenomena (*e.g.*, numerical reasoning, negation and coreference resolution) can be found. For sentence-level perturbations, models are more vulnerable to distraction-based perturbations than directly manipulating syntactic structures. In terms of word-level perturbations, models are similarly vulnerable to different word replacement strategies, among which typo-based perturbations and knowledge-guided perturbations are the most effective attacks.

We hope the above findings can help researchers systematically examine their models against different adversarial attacks, thus also devising new methods to defend against them. Comprehensive analysis of the model robustness report is provided in our website and Appendix A.9.

## 5   Conclusion

We introduce AdvGLUE, a multi-task benchmark to evaluate and analyze the robustness of state-of-the-art language models and robust training methods. We systematically conduct 14 adversarial attacks on GLUE tasks and adopt crowd-sourcing to guarantee the quality and validity of generated adversarial examples. Modern language models perform poorly on AdvGLUE, suggesting that model vulnerabilities to adversarial attacks still remain unsolved. We hope AdvGLUE can serve as a comprehensive and reliable diagnostic benchmark for researchers to further develop robust models.

## Acknowledgments and Disclosure of Funding

We thank the anonymous reviewers for their constructive feedback. We also thank Prof. Sam Bowman, Dr. Adina Williams, Nikita Nangia, Jinfeng Li, and many others for the helpful discussion. We thank Prof. Robin Jia and Yixin Nie for allowing us to incorporate their datasets as part of the evaluation. We thank the SQuAD team for allowing us to use their website template and submission tutorials. This work is partially supported by the NSF grant No.1910100, NSF CNS 20-46726 CAR, the Amazon Research Award.

---

[4]https://github.com/facebookresearch/anli

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
