# OpenReview forum: "Adversarial GLUE: A Multi-Task Benchmark for Robustness Evaluation of Language Models"
_NeurIPS.cc/2021/Track/Datasets_and_Benchmarks/Round2 — NeurIPS 2021 Datasets and Benchmarks Track (Round 2)_

### Official Review · Reviewer_FsxK · 2021-09-10
**Great paper proposing a systematic adversarial benchmark on GLUE**

**Rating:** 8
**Confidence:** 4

**Strengths:**

- It's the first adversarial benchmark that unifies a lot of existing text-attack methods, with a comprehensive coverage over word-level, sentence-level, and human-written adversarial examples. Thus it provides a more systematic picture on model robustness.
- The authors did a thorough filtering via human evaluations, filtering examples where the label is not maintained, and if there's a low agreement among raters. This is a significant contribution as most existing work usually assumes labels don't change when generating adversarial examples, this is also verified by the low attack success rate for almost all methods after filtering in this paper.
- The authors carefully evaluated the output in every step of the entire framework, which would help ensure the high quality of the final benchmark.

**Weaknesses:**

- For data curation, is the human evaluation strictly after automatic filtering? Given the high invalidation rate of the examples, I wonder if relaxing some of the filtering rounds would be more effective. E.g., in Fidelity filtering, you can keep a few more examples (instead of just one) with highest semantic scores, and then go through the human filtering.
- Also, would these steps of filtering cause any biases in the final output? (e.g., the adversarial examples are all too close to the benign examples, and the model could still be vulnerable to examples that are a bit far away)

- For evaluating models, a bit more discussion connecting each model's property with the robustness results would be helpful. E.g., why do some models perform particularly well on certain tasks/certain types of perturbations?

- The example sizes for certain perturbation/task are relatively small, e.g., RTE: C1-C6, QQP: C1, C3, C4, C6, how would the sample size affect the robustness report accuracy? Will they cause a lot of variance?

**Additional Feedback:**

- Can you clarify my question above regarding ALBERT?

- Can you add confidence intervals (e.g., for multiple fine-tuning runs) for each perturbation & task combination, especially when the #samples is small?

**Clarity:**

Yes it's well-written.

Small typo, line 308, "per-trained" -> "pre-trained".

**Correctness:**

To me mostly correct.

One question regarding ALBERT (XXLarge), why do you say it has more parameters and pre-trained with more data / longer time?
- ALBERT (XXLarge) has 235M parameters, comparing to BERT large (334M parameters). ALBERT does have a larger hidden size but the overall model capacity is not larger due to parameter sharing.
- ALBERT is also trained on wikipedia + book corpus, same as BERT.

**Documentation:**

Yes the authors included most of the details.

**Ethics:**

The authors didn't specifically discuss this, but as the paper is trying to improve model's robustness, in general it should help language models become more fair/robust.


**Relation To Prior Work:**

Yes it's well-discussed.

**Summary And Contributions:**

This paper proposes a systematic adversarial benchmark on a subset of GLUE tasks. The paper is well-written, and contributes to the research community greatly in providing a more comprehensive picture on language model robustness.

---

> ### Author Response · Authors · 2021-09-30
> **Thanks for your valuable comments (1/2)**
>
> > **Q1**: For data curation, is the human evaluation strictly after automatic filtering? Given the high invalidation rate of the examples, I wonder if relaxing some of the filtering rounds would be more effective. E.g., in Fidelity filtering, you can keep a few more examples (instead of just one) with highest semantic scores, and then go through the human filtering.
>
> **A1**: Thanks for the interesting question. As discussed in line 41-51 and noted by [1], the goal of the paper is to provide a high-quality adversarial dataset that is accurately and unambiguously annotated and will not fool humans following the definition of adversarial texts. Relaxing the standard may indeed help keep more examples, but will also introduce more noise and lower down both adversarial transferability (effectiveness) and naturalness (fidelity). Thus, we follow the strict standard to guarantee the high quality of our adversarial dataset to effectively evaluate the LMs.
>
>
> >**Q2**: Also, would these steps of filtering cause any biases in the final output? (e.g., the adversarial examples are all too close to the benign examples, and the model could still be vulnerable to examples that are a bit far away)
>
> **A2**: Thanks for the insightful comments.
> First, the goal of adversarial attacks [2,3,4] is to optimize the smallest/stealthy perturbation that misleads the model prediction without fooling humans, which is dangerous if human or standard tools cannot easily identify these abnormal instances and they are sent to language models and potentially do harm. In other words, when the perturbation on texts is too large, these samples are usually easy to detect and defend, as it may dramatically change the semantic meaning and break the grammatical coherence. Thus, in this paper, we follow these principles and launch adversarial attacks under the strict setting to generate adversarial examples and evaluate the model robustness against small adversarial perturbations following the standard adversarial robustness evaluation setup [5,6].
>
> In addition, we note that our AdvGLUE indeed includes human-crafted adversarial examples, such as AdvSQuAD and StressTest, to test model robustness against larger perturbations crafted by humans or rules.
>
>
>
> > **Q3**: For evaluating models, a bit more discussion connecting each model's property with the robustness results would be helpful. E.g., why do some models perform particularly well on certain tasks/certain types of perturbations?
>
> **A3**: Thanks for the valuable suggestions. Large-scale pre-trained language models have intriguing properties and are influenced by a number of factors, including the pre-training data, pre-training strategies, and model architectures. In this paper, we mainly focus on providing a comprehensive and principled understanding of model robustness via diagnostic reports and we will report more details about these model properties in Appendix Section A.6. We hope our work can inspire more future research along this line to understand the intrinsic properties of language models and further improve model robustness.
>
>
> > **Q4**: The example sizes for certain perturbation/task are relatively small, e.g., RTE: C1-C6, QQP: C1, C3, C4, C6, how would the sample size affect the robustness report accuracy? Will they cause a lot of variance? Can you add confidence intervals (e.g., for multiple fine-tuning runs) for each perturbation & task combination, especially when the #samples is small?
>
> **A4**: Thanks for the insightful suggestion. We launched another three fine-tuning runs on BERT (large) and RoBERTa (large) given different random seeds. The evaluation results on GLUE and AdvGLUE can be found in the table below. We can see that although the example sizes of some adversarial perturbations are relatively small, the standard deviation is not high and the mean is consistently aligned with the diagnostic report in Table 5.
>
> |  Model  |	SST-2   |      	MNLI     	| 	RTE	|	QNLI	|      	QQP      	| 	Avg	| Avg (GLUE) |	Avg Δ   |
> |:-------:|:----------:|:---------------------:|:----------:|:----------:|:---------------------:|:----------:|:----------:|:----------:|
> | BERT	| 34.06±0.18 | 27.85±0.41/25.96±0.27 | 36.62±2.41 | 38.88±0.32 | 39.02±0.62/22.0±1.19  | 33.39±0.3  | 85.77±0.12 | 52.37±0.42 |
> | RoBERTa | 44.53±1.22 | 47.17±2.83/39.25±0.59 | 50.0±2.15  | 47.9±0.78  | 51.89±1.01/36.35±1.39 | 45.95±0.38 | 90.39±0.04 | 44.43±0.41 |
>
>
> |  Models | 	C1 	| 	C2 	| 	C3 	| 	C4 	| 	C5 	| 	C6 	| 	C7 	| 	C8 	| 	C9 	| 	C10	| 	C11	|
> |:-------:|:----------:|:----------:|:----------:|:----------:|:----------:|:----------:|:----------:|:----------:|:----------:|:----------:|:----------:|
> | BERT	| 39.81±0.52 | 35.11±0.97 | 37.31±1.98 | 45.46±1.97 | 34.89±0.62 | 41.4±0.8   | 22.62±1.15 | 20.62±0.92 | 22.16±1.83 | 12.61±0.64 | 7.6±1.65   |
> | RoBERTa | 53.59±0.89 | 53.38±0.61 | 55.02±2.07 | 49.11±1.36 | 51.47±0.68 | 46.44±0.52 | 35.21±1.42 | 28.18±1.74 | 35.39±1.75 | 16.41±0.49 | 28.07±2.87 |

---

> > ### Author Response · Authors · 2021-09-30
> > **Thanks for your valuable comments (2/2)**
> >
> > > **Q5**: One question regarding ALBERT (XXLarge), why do you say it has more parameters and pre-trained with more data / longer time?
> >
> > **A5**: Thanks for pointing it out! By “parameters”, we mean ALBERT (XXLarge) has a much larger hidden size (4096) than BERT or RoBERTa (large). The statement is indeed not accurate. We have revised our statements following the suggestions in our revision in Section 4.
> >
> >
> > [1] Bowman, Samuel R. and George E. Dahl. “What Will it Take to Fix Benchmarking in Natural Language Understanding?” NAACL (2021).
> >
> > [2] Carlini, Nicholas and David A. Wagner. “Towards Evaluating the Robustness of Neural Networks.” 2017 IEEE Symposium on Security and Privacy (SP) (2017): 39-57.
> >
> > [3] Madry, Aleksander et al. “Towards Deep Learning Models Resistant to Adversarial Attacks.” ICLR (2018)
> >
> > [4] Jin, Di et al. “Is BERT Really Robust? A Strong Baseline for Natural Language Attack on Text Classification and Entailment.” AAAI (2020).
> >
> > [5] Croce, Francesco, et al.. "Robustbench: a standardized adversarial robustness benchmark." arXiv preprint arXiv:2010.09670 (2020).
> >
> > [6] Carlini, Nicholas et al. “On Evaluating Adversarial Robustness.” ArXiv abs/1902.06705 (2019): n. pag.

---

### Official Review · Reviewer_cpE4 · 2021-09-20
**A Well-Rounded and Exemplified Robustness Evaluation Benchmark for LMs**

**Rating:** 8
**Confidence:** 3

**Strengths:**

- Thorough and comprehensive compilation of adversarial perturbations for language models, which are very carefully compiled and refined

- Each type of adversarial perturbation employed in AdvGLUE is concisely summarized

- Excellent use of examples

- Provided that there exists no reliable benchmark for robustness evaluation under adversarial attacks, this is significant contribution to the community that compliments well the classic GLUE benchmark

- Interesting diagnostic report showcasing how different attacks affect SOTA language models

- Results are very easy to interpret


**Weaknesses:**

- There are too many adversarial perturbation type included in this benchmark, which makes it hard to keep track of all of them. Consider adding a small glossary in the appendix, potentially with an example or two for each type.

**Additional Feedback:**

- In line 43, consider replacing "human" with "humans".

- In section 2, you give a clear motivation for your work, citing that it is hard to guarantee the validity and soundness of existing toolkits and benchmark datasets. However, I believe it would be helpful to relate some of your results from sections 3 and 4 to prior work, namely

 * ALBERT(XXLarge) is the most robust to adversarial attacks
 * Typo-based Perturbations such as text-bugger tend to be the most effective
 * Most SOTA language models evaluated on GLUE are considerably vulnerable to adversarial attacks

In either case, relating these conclusions to prior work can only bolster your overall paper. If these observations are consistent with prior results, then mentioning this would likely give more credibility to your overall benchmark. If these results are new, then emphasizing their novelty will make for an even more interesting paper.

- I believe that motivating your inclusion of the Fleiss Kappa statistical measure would add clarity to your paper, similar to how you justified using the Filter Rate.

**Clarity:**

The paper is clear and nicely exemplified. Results are easy to interpret and follow.

**Correctness:**

The claims made in this paper are substantiated, and the benchmark appears to be very carefully compiled and refined.

**Documentation:**

The data collection, organization and filtering is clearly detailed in the paper. The AdvGLUE official page is also well documented.

**Ethics:**

No ethical concerns.

**Relation To Prior Work:**

The paper is well motivated in the context of previous work.

**Summary And Contributions:**

The authors present Adversarial GLUE (AdvGLUE), a robustness evaluation benchmark for language models that compliments the standard GLUE evaluation with adversarial scenarios. They study and exemplify varying types of perturbations applied to the original GLUE benchmark, and demonstrate how their benchmark is transferrable and can expose vulnerabilities shared between several SOTA language models.

---

> ### Author Response · Authors · 2021-09-30
> **Thanks for your valuable comments**
>
> > **Q1**: There are too many adversarial perturbation types included in this benchmark, which makes it hard to keep track of all of them. Consider adding a small glossary in the appendix, potentially with an example or two for each type.
>
> **A1**: Thanks for the valuable comments. We follow your suggestions and add a glossary of all adversarial attacks considered in AdvGLUE. Specifically, for each adversarial attack, we provide a brief explanation and an adversarial example. We have updated our Appendix and the glossary of adversarial attacks can be found in Table 6 and Table  7 in Appendix Section A.1.
>
>
> >**Q2**: I believe it would be helpful to relate some of your results from sections 3 and 4 to prior work, namely
> > - ALBERT(XXLarge) is the most robust to adversarial attacks
> > - Typo-based Perturbations such as text-bugger tend to be the most effective
> > - Most SOTA language models evaluated on GLUE are considerably vulnerable to adversarial attacks
> >
> > In either case, relating these conclusions to prior work can only bolster your overall paper. If these observations are consistent with prior results, then mentioning this would likely give more credibility to your overall benchmark. If these results are new, then emphasizing their novelty will make for an even more interesting paper.
>
> **A2**: Thanks for the valuable suggestion.
>
> (i) “ALBERT(XXLarge) is the most robust to adversarial attacks”. This observation is aligned with the ANLI leaderboard (without DeBERTa) (https://github.com/facebookresearch/anli), which shows ALBERT (XXLARGE) has the best adversarial robustness to human-crafted adversarial NLI dataset.
>
> (ii) “Typo-based Perturbations such as text-bugger tend to be the most effective.” To the best of our knowledge, we are the first to observe that Typo-based Perturbations such as TextBugger are the most effective and natural adversarial attacks. In this paper, we evaluate the effectiveness and naturalness of 14 state-of-the-art adversarial attack algorithms under the same testbed and evaluation standards. This allows us to perform a comprehensive and fair comparison among existing attacks and draw such interesting conclusions.
>
> (iii) “Most SOTA language models evaluated on GLUE are considerably vulnerable to adversarial attacks”. To the best of our knowledge, we are the first to observe that SOTA language models are also vulnerable to adversarial attacks. Most adversarial attacks [1,2,3] focus on evaluating the robustness of BERT and RoBERTa language models. In this paper, we perform a principled benchmark over state-of-the-art language models and reach such conclusions.
>
> We also revised our paper accordingly in Section 4.
>
>
>
> > **Q3**: I believe that motivating your inclusion of the Fleiss Kappa statistical measure would add clarity to your paper, similar to how you justified using the Filter Rate.
>
> **A3**: Thanks for the insightful comments. Fleiss Kappa is a widely used metric in existing datasets (e.g., SNLI, ANLI, and FEVER [4,5,6]) to measure the inter-annotator agreement rate on the collected dataset. Fleiss Kappa between 0.4 and 0.6 is considered as moderate agreement and between 0.6 and 0.8 as substantial agreement. The inter-annotator agreement rates of most high-quality datasets fall into these two intervals. In this paper, we follow the standard protocol and report Fleiss Kappa and Curated Fleiss Kappa to analyze the inter-annotator agreement rate on the collected adversarial dataset before and after curation to reflect the ambiguity of generated examples. We also updated our paper about the motivation and description of Fleiss Kappa in Section 3.4.
>
> [1] Jin, Di et al. “Is BERT Really Robust? A Strong Baseline for Natural Language Attack on Text Classification and Entailment.” AAAI (2020).
>
> [2] Li, Linyang et al. “BERT-ATTACK: Adversarial Attack against BERT Using BERT.” EMNLP (2021)
>
> [3] Li, Dianqi et al. “Contextualized Perturbation for Textual Adversarial Attack.” NAACL (2021).
>
> [4] Bowman, Samuel R. et al. “A large annotated corpus for learning natural language inference.” EMNLP (2015).
>
> [5] Nie, Yixin et al. “Adversarial NLI: A New Benchmark for Natural Language Understanding.” ACL (2020)
>
> [6] Thorne, James et al. “FEVER: a Large-scale Dataset for Fact Extraction and VERification.” NAACL (2018).

---

### Official Review · Reviewer_fGCh · 2021-09-20
**Very Insightful Paper**

**Rating:** 8
**Confidence:** 4
**Clarity:** The paper is well written.

**Strengths:**

- The paper proposes a principle and comprehensive adversarial robustness benchmark for NLU, which is very useful for research in both NLU and adversarial robustness in NLP.
- The paper points out the weakness of adversarial attack methods with human checks in NLP, which is novel and very insightful. The finding could inspire future research on related areas.
- The benchmark shows good accessibility. Models can be evaluated on both GLUE and the proposed AdvGLUE with one-time training.

**Weaknesses:**

- It will be better if more pre-trained large-scaled language models are evaluated and presented, but I don't think it's a big issue.

**Additional Feedback:**

Line 254: servers -> serves

**Correctness:**

I don't find a correctness problem. The datasets are constructed in a sound way and the evaluation methods are reasonable.

**Documentation:**

Yes.

**Ethics:**

I don't find an ethical concern.

**Relation To Prior Work:**

Yes, it is.

**Summary And Contributions:**

The paper introduces a new adversarial robustness benchmark for natural language understanding (NLU) tasks called AdvGLUE. The proposed benchmark is based on the widely-used GLUE benchmark and practitioners can be evaluated on both with only one-time training.

14 state-of-the-art textual adversarial attack methods, including word level, sentence level, and human-crafted attacks are used to generate the adversarial examples on original GLUE dev sets, making the benchmark quite comprehensive. The authors evaluate several state-of-the-art large-scaled language models and defense methods and show that these models are quite weak in robustness.

As another very meaningful contribution, the authors make human checks on the adversarial examples and find that most generated adversarial attacks are invalid samples. After human curation, the attack success rate significantly drops than before.

In sum, this paper shows that there is a long way to go for both adversarial defense and attack methods in NLP. The benchmark and the findings could shed light on future research directions in NLP.

---

> ### Author Response · Authors · 2021-09-30
> **Thanks for your valuable comments**
>
> We thank the reviewer for the insightful comments and suggestions, and we provide the response and additional evaluations below.
>
> > **Q1**: It will be better if more pre-trained large-scale language models are evaluated and presented, but I don't think it's a big issue.
>
>
> **A1**: Thanks for the valuable suggestion.  We follow your suggestion and evaluate more pre-trained large-scale language models. Specifically, we fine-tune another two state-of-the-art language models: DeBERTa (large) and T5 (large). The fine-tuning details and hyper-parameters can be found in Appendix A.6. We evaluate these models on both the benign GLUE dev set and our AdvGLUE, which are shown in the tables below. We can see that DeBERTa (large) achieves the highest robust accuracy on AdvGLUE. We also updated Table 4 and 5 in our paper in Section 4.
>
> |  	Model  	| SST-2 | 	MNLI	|  RTE  |  QNLI | 	QQP 	|  Avg  | Avg (GLUE) | Avg Δ |
> |:---------------:|:-----:|:-----------:|:-----:|:-----:|:-----------:|:-----:|:----------:|:-----:|
> | T5 (large)  	| 60.56 | 48.43/38.98 | 62.83 | 57.64 | 63.03/55.68 | 56.82 | 90.39  	| 33.57 |
> | DeBERTa (large) | 57.89 | 58.36/52.46 | 78.95 | 57.85 | 60.43/47.98 | 60.86 | 92.67  	| 31.81 |
>
>
>
> |  	Models 	|  C1   |  C2   |  C3   |  C4   |   C5  |   C6  |   C7  |   C8  |  C9   |  C10  |  C11  |
> |:---------------:|:-----:|:-----:|:-----:|:-----:|:-----:|:-----:|:-----:|:-----:|:-----:|:-----:|:-----:|
> | T5 (large)  	| 60.04 | 67.94 | 64.60 | 59.84 | 58.50 | 50.54 | 42.20 | 69.02 | 23.20 | 17.10 | 52.63 |
> | DeBERTa (large) | 65.07 | 74.87 | 68.02 | 65.30 | 62.54 | 57.41 | 47.22 | 45.08 | 52.06 | 22.80 | 54.39 |

---

### Author Response · Authors · 2021-09-30
**General Response**

We thank all the reviewers for their time and valuable suggestions. Based on the reviews, we have corrected several typos and made the illustration more clear. We also include more experimental results.

Specifically, we made the following revisions:

1. We revised the typos pointed out by the reviewers in Line 43, 253, and 313.
2. We added the evaluation results of another two state-of-the-art pre-trained language models T5 and DeBERTa in Table 4, 5, and 9 in Section 4.
3. We added a glossary of adversarial attacks considered in AdvGLUE with explanations and examples in Table 6 and 7 in Appendix A.1.
4. We related some of our observations to prior work in Section 4 to confirm the conclusion.
5. We clarified the motivation and details of Fleiss Kappa in Section 3.4.
6. We added more training details in Appendix A.6.
7. We clarified some experimental analysis in Section 4.

All of our revisions are updated in OpenReview and highlighted in blue. Thank you!

---

### Decision · Program_Chairs · 2021-10-09

**Decision:**

Accept

**Comment:**

This is an interesting submission and all reviewers recommend acceptance, hence I also recommend accepting the paper.

My only concern is whether adversarially constructed test sets are truly harder than non-adversarial distribution shifts. For instance, the Swag benchmark was constructed via adversarial filtering and current models achieved only low performance scores, but two months after the release of the Swag paper, the BERT paper achieved the same performance on Swag as a human expert. Similarly, models have made quick progress on the adversarially-filtered ImageNet-A test set (e.g., see Figure 4 in https://arxiv.org/abs/2007.00644). I would be curious to hear the authors' perspective on this point (and it may be a valuable addition for readers of the paper as well).

---

> ### Author Response · Authors · 2021-10-14
> **Thanks for your valuable comments**
>
> We thank the program chairs for the interesting and insightful questions.
>
> Regarding the hardness of adversarially constructed test sets, both our current and existing studies show that the adversarially constructed data is very successful in terms of attacking the language models and they also have high transferability across different models. Thus, we believe such attacks are indeed important tests for evaluating the robustness of language models and so far there is no successful defenses against them yet, which is a very challenging open problem not only in natural language processing but also other general domains such as computer vision.
>
> In addition, multiple rounds of filterings in terms of fidelity and transferability ensure the constructed samples can unveil the adversarial vulnerabilities shared across models and attack unseen language models.
>
> Moreover, AdvGLUE consists of a comprehensive attack taxonomy of 14 state-of-the-art attacks to benchmark the model robustness based on different levels of attacks, which is flexible enough to estimate the model robustness on different levels.
>
> Finally, when models make progress on the robustness, our benchmark can still serve as a standard and fair leaderboard to compare the relative robustness between existing models, which is one of the main goals of AdvGLUE, and we believe it will largely boost our understandings about the robustness and properties of different language models.